# Long-Term Care Insurance Pilot Programme in China: Policy Evaluation and Optimization Options—Taking the Pilot Programme in the Northeast of China as an Example

**DOI:** 10.3390/ijerph19074298

**Published:** 2022-04-03

**Authors:** Ye Han, Tong Shen

**Affiliations:** Department of Labor and Social Security, Jilin University, Changchun 130012, China; shentong20@mails.jlu.edu.cn

**Keywords:** long-term care insurance (LTCI), policy pilot programme, policy evaluation, optimization options

## Abstract

China’s population is ageing rapidly and the increase in life expectancy is accompanied by a loss of capability with advancing age, especially in the Northeast. This study adopts qualitative research methods to analyze the overall status and problems of China’s LTCI policy pilots. Taking four LTCI pilot cities in three northeastern provinces as samples, we used purposive sampling to recruit 10 beneficiaries and providers of LTCI in nursing homes of different kinds, as well as 2 operators (Medical Insurance Bureau staff) for semi-structured in-depth interviews. We developed a social welfare policy analysis framework based on Gilbert’s framework, designed interview outlines and conducted a thematic analysis of the interview data along five dimensions: allocation base, type of provision, delivery strategy, finance mode, and external environment. The results of the research indicate that the coverage of the system is narrow and that disability assessment criteria are fragmented; that the substance of service provision is lacking, both in terms of precision and dynamic adjustment mechanisms; that socialized care synergy cannot be achieved, informal care lacking policy support; that there is an over-reliance on medical insurance funds and that unfair financing standards are applied; and that economic and social development is insufficient to cope with ageing needs and uncertain risks. Accordingly, this research proposes several optimization options to promote the full establishment of LTCI.

## 1. Introduction

According to the data of the Seventh National Census, the population aged 60 or older in China is 264.02 million, accounting for 18.70% of the total population, of which 190.64 million are aged 65 or older, accounting for 13.50% of the total population. According to a report on the data of the Sixth National Census, the proportion of the population aged 60 or older has increased by 5.44 percentage points. Compared with other countries, the phenomenon of increased ageing, accompanied by empty nesting, higher life expectancy and loss of capacity, is particularly prominent in China. According to research conducted by Peking University, China’s disabled elderly population reached 52.71 million in 2020 and is projected to account for more than 57% of the total disabled population by 2030, with a total of more than 77,656,800 [1]. Nowadays, it has become indisputable that the family care function has weakened under the dual influence of the miniaturization of family size and the increase in female employment [2]. How to cope with the rapidly increasing demand for care of the disabled elderly and the high cost of care has become a real problem that needs to be solved.

Currently, promoting long-term care (LTC) security systems through the establishment of social insurance systems is mainstream in countries looking to address the social risks of care. Typical countries include the Netherlands, Germany, Japan, and South Korea. The LTCI systems in these four countries were established under approximately the same socioeconomic context: once the transition to an ageing society was underway, it was recognized that the population of the country was ageing rapidly, that the degree of childlessness was increasing along with it, and that the demand for disability care for the elderly was increasing significantly. When the systems were established, the proportion of elderly aged 65 or older was 9.8% in the Netherlands [3] and reached 15.5%, 17.4%, and 10.3% in Germany, Japan, and South Korea [4], respectively. Traditionally, families have taken responsibility for caring for the elderly. However, with industrialization and urbanization and increased female employment rates, families have become smaller and more dispersed. Family caregiving capacity has been gradually weakened. The demand for care spills over and evolves into social risk. Therefore, the aforementioned countries have established LTCI systems to provide an institutional response to these societal needs. Although economic development has accumulated a strong economic foundation in these countries (the GDP per capita of all the four countries exceeded USD 20,000 at the time of establishment), due to the lack of care resources and compensation mechanisms, the elderly have to seek the support of medical services and social assistance, which has led to a serious waste of medical resources and the overburdening of government finances [5]. To address this issue, such countries have developed laws to legally establish LTCI systems. On 14 December 1967, the parliament of the Netherlands passed the Act on Special Medical Expenses, and the world’s first social insurance-based LTCI system was established [6]; on 26 May 1994, after nearly two decades of discussion on the model for establishing care guarantees, the German parliament passed the Long-term Care Insurance Act, which added LTCI as the fifth pillar of the German social insurance system [7]; the Care Insurance Act was enacted in Japan in 2000 to formally implement the LTCI system [8]; in South Korea, after three-phase pilot programmes, the Long-Term Care Insurance for the Elderly Act officially came into effect on 1 July 2008, establishing the LTCI system [9].

Despite solid legal support, sustainability problems such as insufficient service supply and high fund deficits have still arisen during the implementation of LTCI systems in these countries, the systems affected by uncertain risks and operating costs. After various adjustments and reforms, the results are still not satisfactory: because of the overemphasis on state intervention and government supply, users and families lack a sense of responsibility. The LTCI system in the Netherlands is inefficient and heavily dependent on welfare. Huge financial pressures have forced the Netherlands to cut informal care. Due to the imperfect development of the informal care system and support policies, the care supply in the Netherlands is seriously insufficient [6,10]. Meanwhile, affected by the age structure of the population, the German LTCI system has experienced a serious deficit in insurance premiums since 2000. The insurance premium rate has been raised several times, but it is still unable to meet the ageing of the population and the rigidity of the system [7]. Moreover, as a result of a lack of informal caregivers and an ill-designed system, the institutional principle of “home-based care prior to institutional care” cannot be implemented [11]. The LTCI system in Japan only covers citizens over 40, which not only violates social equity but also limits the scale of fundraising. At the same time, although treatment is already too generous, the proportion of personal benefits is small, the system being prone to moral hazard and waste of care resources. In addition, the fact that local governments are the main bodies responsible for operations is not conducive to the overall planning by central government, leaving the system prone to the risk of fragmentation [12]. In South Korea, the system has a narrow coverage due to the harsh conditions respecting entitlements, and in 2010 only 3.1% of the total elderly population was covered by the system [9,13].

Different from the legislation of Western countries, China has always had a tradition of testing the rationality, feasibility, and scientificity of policy plans through small-scale pilot programmes before the formal system is introduced [14]. In a pilot programme it is easy to control various risks in the implementation of the system and the decision-making process, so that the scientificity and controllability of risk prevention and control can be improved. When adjusting policies based on accumulated experience, it can also disperse trial and error costs [15]. After fully learning from the experience of typical countries, China has started a pilot programme of an LTCI system. In June 2016, the Ministry of Human Resources and Social Security issued the *Guiding Opinions on Launching the Pilot Programme of the Long-Term Care Insurance System* and decided to conduct the first pilot programme in 15 cities, including Qiqihar, Changchun, and Qingdao. In September 2020, the National Medical Insurance Administration and the Ministry of Finance jointly issued the *Guiding Opinions on Expanding the Pilot Programme of the Long-Term Care Insurance System* and decided to add 14 pilot cities, including Panjin, Tianjin, and Huhhot across the country. The goal is to build a framework for the LTCI system that adapts to the level of China’s economic development and the trend of ageing in the new development pattern, and eventually to establish a sound multi-level LTCI protection system that meets the diversified needs of the public. Up to now, all pilot cities have issued relevant policy guidance documents, and the pilot work has been fully implemented at the policy-guidance level.

Many studies in China have assessed the feasibility and operational model of the LTCI from an economic perspective with a quantitative approach or have provided a characteristic analysis and experience-based summary of local practice in a pilot city. These studies tend to focus on a particular institutional element or a unique pilot city. While relevant and scientific, these studies easily generalize and deviate from the original intent of the system. Therefore, it is only by analyzing the guiding elements of the LTCI policy objectives, value concepts, and theoretical structure that we can attain a holistic grasp of it. Policy analysis is the description and explanation of policy values, theories, and specific policy processes guided by a certain theoretical framework. It is intended to help policy decision-makers and implementers to improve the process of policy formulation and implementation [16]. The dimensions of policy analysis are mainly divided into empirical and normative dimensions. Herbert Alexander Simon has noted that any decision contains both factual and value factors. Empirical analysis understands the causes and consequences of policy issues by observing, describing, and evaluating things, events, and relationships related to policy processes. It is an analysis of the factual elements of policy. Normative analysis evaluates alternatives through a set of criteria and ultimately selects the most satisfactory policy option. It is an analysis of policy value factors. The development and selection of evaluation criteria are influenced by subjective factors, such as the preferences, habits of mind, and values of the evaluator [17].

Different scholars have constructed various policy analysis frameworks based on value judgments or research perspectives. Most belong to normative analyses, wherein policy options are analyzed by developing evaluation criteria. The results of these policy analyses are more influenced by the analysts’ own culture, values, and other factors. In contrast, in the field of social welfare policy, policy output analysis distinguishes the basic components of a policy programme and examines these values and theoretical structures associated with the choice [18]. It is not affected by different social contexts or the ideology of the analyst. The results of its analysis have relative objectivity and policy comparison is feasible. It is more applicable to the comparison, evaluation, and problem analysis of LTCI pilot policies. The social welfare policy analysis framework proposed by the American scholar Neil Gilbert in *Dimensions of Social Welfare Policy* in 1986 is the representative framework for policy output analysis. By integrating the complementary elements of social welfare policy to form an independent and complete analysis system, he provides a theoretical perspective for the analysis of social welfare policy. He summarized four basic dimensions of social welfare policy: allocation base, type of provision, delivery strategy, and finance mode [18]. The allocation base consists of guidelines for determining who is eligible for social welfare; the types of provision refer to the choice of the form and substance of social welfare, which represents the form and substance of welfare that beneficiaries can enjoy; the delivery strategy refers to certain organizational arrangements of social welfare among providers and recipients of service, which solves the problem of how welfare objects obtain benefits; the finance mode refers to the financing mechanism of social welfare, which solves the problem of the sources and financing method of social welfare funds. However, the design, establishment, operation, and future development of the social welfare system cannot be separated from the social reality. The socioeconomic and political environment, along with uncontrollable contextual factors, will affect the welfare system. Therefore, objectively analyzing the external environment in which the system operates is of practical significance. This study innovates the analytical framework for social welfare policy and defines the external environment as an element for ever-changing social welfare system, being a fifth dimension for evaluating social welfare policy.

This study uses a policy analysis approach to review the philosophy and direction implied by the development of the LTCI system. We use the LTCI policies promulgated by the pilot cities, take the pilot programmes of the four pilot cities in the three northeastern provinces as a sample, and treat the pilot documents officially issued by the human resources and social services authorities or health insurance authorities in such cities as the policy texts. We analyze the overall status and common features of the LTCI pilot programmes along five dimensions: allocation base, type of provision, delivery strategy, finance mode, and external environment. The aim of the study is to explore the common problems that have arisen in the current policy practices and to provide useful references for further expansion of the pilot programme and the full establishment of LTCI.

## 2. Materials and Methods

### 2.1. Design

A fixed questionnaire may overlook respondents’ indelible personal experience, subjective feelings, and policy reflections on LTCI. To obtain more direct and vivid first-hand information, this study designed interview questions along five dimensions. The qualitative research method was used to collect in-depth interview data to examine the subjective experiences of the disabled elderly (i.e., LTCI beneficiaries), the administrators, caregivers and medical staff of nursing homes (i.e., LTCI service providers), and the staff of the health insurance bureau (i.e., LTCI operators). An in-depth interview is a type of unstructured, direct, deep, and one-to-one interview, which is an appropriate way to collect data on the potential motives, experiences, attitudes, and emotions of the respondents regarding a certain issue [19]. This study used in-depth interviews to collect the real feelings and behavioral choices of LTCI policy participants, which data can be used to summarize the common problems in the implementation of LTCI policies and thus provide useful references for policy optimization.

This study was conducted in four LTCI pilot cities in the three northeastern provinces of China: Qiqihar in Heilongjiang Province, Changchun and Tonghua in Jilin Province, and Panjin in Liaoning Province. The population in the three northeastern provinces is ageing at a faster rate than the national average. The elderly situation is more severe, and the demand for disability care is significant. Industrial structural transformation has rendered the old industrial base relatively backward in terms of economic development. Coupled with the traditional concept of old age and the mass exodus of the population, the three northeastern provinces are in urgent need of a subsidized LTCI system and service supply of a care guarantee system.

### 2.2. Participants

We recruited participants through the staff of the Association of Senior Care Services Industry. The Association of Senior Care Services Industry is an organization that undertakes consultation, business guidance, and training for senior care institutions, cooperates with the government, and reflects the demands of its members. With the help of the Association, we selected one nursing home in each of the four pilot cities. Participants were recruited through purposive sampling, considering specific criteria related to the research objectives [20]. The criteria were as follows: (1) the nursing home was the designated institution for LTCI; (2) clear cognition of the participant and no communication barrier with the researchers; (3) the elderly individuals needed to have LTC needs; and (4) informed consent was given and participation was voluntary. In order to fully reflect the operation of LTCI in nursing homes of different natures, the types of nursing homes we choose covered three categories: public non-profit, private non-profit, and private for-profit. Affected by the closed-loop management of nursing homes due to the COVID-19 epidemic and the need to find fluently expressive and severely disabled elderly with LTC needs, we eventually recruited a total of 10 participants, including 4 managers, 2 caregivers, 1 medical staff member and 3 elderly individuals with disabilities. In addition, we recruited two Medicare staff who provided relevant information about the status and future development of LTCI operations. Table 1 describes the demographic data and basic information of the participants in the study.

The respondents were five males and seven females. The elderly were aged between 78 and 86 years and all were severely disabled. They had clear cognition and no communication barriers. All four nursing homes were LTCI-designated institutions, including two private for-profit institutions, one private non-profit institution, and one public non-profit institution.

### 2.3. Data Collection

The interviews took place in August and December 2021. The authors completed all one-on-one in-depth interviews in Mandarin Chinese, each ranging from 30–45 min. Before the interviews, we contacted the participants and provided general information related to our research (including the purpose, theme, process, and plan) and the interview procedure. After obtaining the consent of the interviewees, the interviewers recorded the dialogues through on-site audio-recording and note-taking to collect raw data. The notes included the core content of the interview and the physical and mental state of the interviewee. Due to the impact of the COVID-19 epidemic, parts of the interviews were conducted within the nursing homes and parts of the interviews were conducted in an online connected format. The authors are researchers who have received training in qualitative methods and interview techniques. We designed a semi-structured interview schedule according to the purpose of the research. The interview schedule asked two to three questions about each of the following subjects: the allocation base, the type of provision, the delivery strategy, the finance mode, and the external environment of LTCI policies. Table 2 details the content of the interview schedule.

### 2.4. Data Analysis

Given the purpose of the research, we used thematic analysis methods employed in qualitative research to analyze the data. The thematic analysis method is a descriptive method of “identifying, analyzing and reporting patterns (themes) in data” [21] which flexibly reduces data and makes connections with other data analysis methods [22]. By applying this method, researchers can gain a better and deeper understanding of the beneficiaries, providers, and operators’ attitudes, visions, feelings, and reflections on LTCI and extract core themes therefrom. We used the NVivo 11.0 (QSR International Pty Ltd., Burlington, MA) software packagefor data analysis. First, we converted the recorded data and notes into text, read the raw data repeatedly to familiarize ourselves with all the dimensions of the data, and then extracted meaningful statements to generate initial codes, such as “the elderly in rural areas and migrant workers are not guaranteed” and “the elderly are not recognized”. Second, when all the data had been initially coded and collated, we focused the analysis at the broader level of themes by collating all the relevant codes into themes such as “difficulty in service delivery” and “lack of policy support for informal care”, then reviewed the themes and checked that they related properly to the initial codes before defining the themes. Finally, to ensure the reliability of the analysis, we returned the initial codes and thematic codes to the respondents and invited suggestions for these codes.

### 2.5. Ethical Considerations

During the in-depth interview process, our research strictly followed the procedures of informed consent, non-harm, and confidentiality of participants. On the one hand, before the interview, we explained the purpose and use of the interview as well as the recording requirements in detail, and all participants gave signed informed consent. Due to the relatively fragile physical and mental condition of the disabled elderly individuals, we asked the staff of the nursing home to communicate fully with them before the interview began. For confidentiality, all identifying information was anonymized during the transcription and translation process. Any sensitive material from the interviews was technically processed.

## 3. Results

### 3.1. The Specific Performance of the LTCI Pilot Programmes under the Five-Dimensional Framework of Policy Evaluation

#### 3.1.1. Allocation Base: Coverage Based Principally on the Principles of “Selective Allocation + Diagnostic Differentiation”

The allocation base addresses the question of how to determine the covered persons. Generally, under the benefits based on the principle of selectivity, the recipients are selected subject to the conditions of benefit payout. For the social insurance system, the target population and the conditions for benefit payout directly determine the scope and number of the covered persons. In practice, LTCI policies in nearly 70% of the pilot cities only cover participants in Urban Employee Basic Medical Insurance (UEBMI). For example, in Qiqihar and Panjin it is stipulated that the covered persons in the pilot phase shall first be urban employees. Only in a few cities has the coverage of the LTCI been moderately expanded, such as in Changchun and Tonghua, where it is required that urban employees (including retirees) or urban–rural residents covered by basic medical insurance shall be automatically covered by urban employees’ or urban–rural residents’ LTCI. However, in practice, non-urban residents are still not permitted to participate in the LTCI at their own discretion. Comparing the two pilot policies, it was found that the expansion of the pilot programme did not extend the scope of the insured, and the policy coverage was still very narrow.

On the other hand, Gilbert further refined the allocative principles into attributed need, compensation, diagnostic differentiation, and means-tested need [18]. The benefit payment of our LTCI policy is built on an assessment of the degree of disability of the benefit applicant. It is a diagnostic differentiation type of eligibility review, which makes professional judgments on the special needs of benefit applicants. Therefore, the degree of perfection of disability assessment criteria directly determines the coverage of benefit recipients. At present, in most pilot cities, the Barthel scale is used to determine the severely disabled elderly as the benefit recipients. For example, in Changchun, Tonghua, Qiqihar, and Panjin, it is stipulated that LTCI focuses on addressing the basic nursing care needs of severely disabled persons. The insurance does not cover the elderly with dementia. There are only six pilot cities in China where care for the elderly with dementia is provided. For example, in Qingdao, long-term, daytime, and temporary care are provided for the elderly with severe dementia based on an evaluation using a professional dementia assessment scale; in Shanghai, cognitive bed units are set up in senior care institutions for the elderly with severe dementia based on a comprehensive scale; institutional care and home-based care are provided for the elderly with severe and moderate dementia in Guangzhou, and for the elderly with severe disability due to dementia in Chengdu [23]. On 3 August 2021, the first set of unified standardized disability-level assessment criteria at a national level, i.e., the *Long-term Care Disability Level Assessment Criteria (Trial)* (the “Assessment Criteria”) was issued, covering comprehensive indicators of ability assessment, such as the ability to carry out activities of daily living, cognitive ability, perceptual and communication abilities, and providing strong support for the balanced benefit and system equity of the LTCI system.

#### 3.1.2. Types of Provision: Institutional Centralized Care or Home-Based Care as the Main Form of Basic Care Services

Traditionally, types of welfare provision are mainly divided into provision in cash and provision in kind. Since China’s resources and provision bodies for care services are insufficient and the needs of most of the disabled elderly are relatively concentrated and clear, it is more appropriate for the LTCI system to adopt the service supply method in the form of in-kind provision. In practice, services in the forms of institutional care and home-based care are provided in most pilot cities. For example, Qiqihar provides three forms of services: medical institutional care, nursing institutional care, and home-based care. Institutional care and home-based care are provided in Panjin for the severely disabled. Changchun is exploring the establishment of a home bed care model for the elderly. In addition, Nanning, a pilot city in 2020, also provides care services for people with severe disabilities living in other places so that they can receive care services at their place of residence after concluding an agreement with a designated care service provider.

In terms of the substance of service provision, two main types of services, i.e., medical care and daytime care, are provided in the pilot cities. For instance, the care services currently provided in Changchun, Tonghua, Qiqihar, and Panjin are mainly focused on basic daytime care services and basic medical care services for the severely disabled. The service provision in Qingdao, which took the lead in pilot exploration, is rich and comprehensive. There are not only basic life care and medical care services for people with complete disability and severe dementia, but also training for maintaining body functions for semi-disabled individuals and those with mild-to-moderate dementia. The expansion of the pilot programme did not lead to a greater expansion in terms of the form and substance of services. The provision of services has yet to meet the diverse and multi-level care needs.

#### 3.1.3. Delivery Strategy: State-Led + Social Force Participation as the Main Service Delivery Mechanism

The delivery system refers to the organizational arrangements that exist among service providers and between service providers and consumers in the context of the local community [18]. Service providers can be professionals, self-help associations, or public and private agencies; they can also be formal care providers or informal caregivers, such as family members, neighbours, and friends. Historical practice shows that making the government or the market the sole provider of social welfare will lead to a double failure of the government and the market. Only by focusing on the strength of society, communities, families, and individuals can national social welfare be truly maximized [24]. In this process, not only should the state’s responsibility for basic welfare provision be stressed, the role of market mechanisms should also be leveraged. In terms of service providers, each pilot city has identified qualified medical and nursing institutions as designated LTCI institutions according to their own standards. For example, Qiqihar, Changchun, Tonghua, and Panjin all stipulate that qualified medical and nursing institutions and senior care institutions be invoked as care service providers. The first batch of pilot cities, including Suzhou and Shangrao, also included community senior care service institutions, nursing homes (stations), outpatient departments, community health care centers, and other grass-roots medical and nursing institutions as service providers, making full use of grass-roots medical and care resources to strengthen service supply. The expansion of the pilot programme has attracted more social forces to become involved in service supply, and the total number of providers has increased. However, the entities implementing the policies are still led by the government, with insufficient participation of the market and civil society.

#### 3.1.4. Finance Mode: A Social Insurance Model Financed Mainly by the Medicare Insurance Pool

The funding mechanism is a key component of the sustainable operation of a social insurance system. In order to ensure further expansion and the full establishment of LTCI, the funding mechanism needs to be fair, independent, and sustainable. On the one hand, funding criteria affect the fairness of the LTCI system. At present, pilot cities have explored three types of funding: quota funding, funding by a fixed proportion, and gradated funding standards for different groups. For example, Qiqihar stipulates that the fundraising standard is CNY 100/person/year (including CNY 50 for organizations and CNY 50 for individuals), which is a form of quota financing. Panjin distinguishes the contribution bases of on-the-job employees, flexible employees, and retirees. On-the-job employees and flexible employees are funded at 0.4% of the UEBMI payment base, and retirees are funded at 0.4% of the UEBMI personal account, which belongs to the proportion of different payment bases. Changchun stipulates that urban employee LTCI groups raise funds at 0.2% of the UEBMI payment base, while urban–rural resident LTCI groups are to be funded at a standard fixed amount of CNY 12. Tonghua’s payment level is slightly lower than that of Changchun. The financing ratio is 0.15% of the UEBMI payment base. Both cities provide gradient financing for groups of diverse regions and identities.

On the other hand, funding channels affect the sustainability and independence of funding sources. At present, each pilot city has explored two kinds of funding: single-channel and multi-channel. Most of the pilot cities have two or more financing channels, focusing mainly on the transfer of medical insurance funds and the adjustment of the structure of medical insurance unified accounts. For instance, Panjin stipulates that fundraising is mainly based on the payment of organizations and individuals. In the initial stage, the organization contribution is deducted from the medical insurance payment or the accumulated balance of the overall fund, and the individual contribution is taken from the medical insurance personal account. Qiqihar mainly raises funds by optimizing the structure of employees’ medical insurance unified accounts and transferring the balance of the overall fund. Individual contributions and appropriate supplements to finances are also included. Changchun and Tonghua raise funds through various channels, such as individual contributions, organizational contributions, financial subsidies, and social donations. In addition, there are still several cities such as Ningbo and Guangzhou that only rely on the transfer of medical insurance funds to raise funds. Comparing the policies of the two pilot programmes, the expansion of the pilot programme has improved the financing standards to a certain extent, and the financing channels have been previously expanded. Diversified financing mechanisms have been basically established. Financial aid has played a more important role.

#### 3.1.5. External Environment: The Social Characteristics of “Growing Old before Getting Rich” Combined with Public Health Emergencies in the Context of the New Economic Normal

In 2015, China’s economy entered a new normal development stage: the economic growth rate shifted from high speed to medium speed, the economic structure was optimized and upgraded, and the economic growth momentum shifted from being factor-driven and investment-driven to being innovation-driven. The slowdown in economic growth constrained the government’s investment in social security. Industrial restructuring triggered production and operational difficulties for some enterprises, which, in turn, affected the stability of insurance contributions. Since LTCI has been attached to the pool of medical insurance funds since its establishment, the new features of economic development have had a certain impact on institutional financing. In the context of the recent economic normal, China has the typical characteristic of “ageing before the rich” in dealing with the ageing population. According to the seventh national census data, the population aged 60 years or older in Qiqihar accounts for 24.13% of the total population, while this demographic represents 23.06%, 25.90%, and 22.67% of the total populations in Changchun, Tonghua, and Panjin, respectively [25]. The four cities have higher levels of ageing and disability care needs. However, in terms of economic development level, the figures for Changchun’s and Panjin’s GDP per capita in 2020 were CNY 77,634 [26] and CNY 93,804 [27], respectively; these were basically the same as the national per capita level of CNY 72,447, while the figures for Qiqihar and Tonghua were only CNY 24,273 [28] and CNY 34,338 [29], respectively—far below the national average standard. The economic strength with which to cope with the ageing population is relatively weak. Further, the new economic normal has been accompanied by the promotion of supply-side structural reform. As a representative of the old industrial bases, Northeast China has long relied on resource-based industries and heavy industries. With the transformation of industrial structure, resource depletion, and diminishing marginal rewards of factors, traditional industries are not able to support regional economic growth. The three northeastern provinces are facing structural difficulties with respect to employment, which puts greater pressure on social insurance collections and payments.

It is important to emphasize that the combination of “growing old before getting rich” and public health emergencies have further worsened the external environment for the LTCI system pilot. On the one hand, COVID-19 was spreading globally in early 2020. Depending on the World Health Organization weekly report, as of 19 December 2021, more than 273 million confirmed cases of COVID-19 and more than 5.3 million deaths had been reported globally [30]. Although the Chinese people are united and actively fighting the pandemic and with a control blockade are setting an example to people all over the world, the COVID-19 pandemic still introduces great uncertainty into the LTCI system, due to its high pathogenicity, high infection rates, extremely rapid transmission, and constant mutation. On the other hand, the elderly constitute a high-risk group for virus susceptibility and the most vulnerable to the outbreak. According to foreign media reports, in some european countries, some elderly were abandoned by nursing homes [31] and confirmed that there were elderly individuals who were not even given access to hospital care [32]. After the outbreak, the Chinese government supported the prevention and control of the epidemic by investing most of its financial resources, adjusting health insurance policies and even allocating health insurance funds. Despite the fact that it fully reflects humanitarianism and protects the right to life and health of the elderly, it also restricted the promotion of the LTCI system to a certain extent.

### 3.2. The Main Problems in the Evaluation of China’s LTCI Pilot Policies

#### 3.2.1. Allocation Base: Narrow System Coverage and Fragmented Disability Assessment Criteria

As a social insurance system that aims to alleviate LTC problems through collective mutual assistance, the LTCI policies must first ensure that all members of society are eligible for coverage. At present, the pilot policy cannot cover all members of society—rural residents, migrant workers, and employees in new forms of employment have not received due protection. Limiting insurance on the basis of occupation or region violates social justice and will cause social conflicts, affecting the sustainability of the LTCI fund.


*At present, the elderly in our institution who enjoy LTCI are severely disabled. Now the LTCI in our province is mainly aimed at UEBMI enrollees, and I think the coverage is not very wide. I hope that in addition to them, the government can give some subsidies to the new rural cooperative medical insurance participants, the Urban-Rural Resident Basic Medical Insurance (URRBMI) enrollees, and even those who are not able to participate in any insurance. Because the proportion of the disabled elderly is very high now, those who cannot obtain insurance, such as rural elderly and migrant workers, are precisely low-income families in difficulty, and LTCI reimbursement is the most critical of them.*
(P-10)


*Severely disabled elderly and senior elderly are eligible for LTCI at our institution. Now the policy requires that those who enjoy the treatment must be registered in City C, which means that a large number of elderly who have relocated with their children cannot enjoy subsidies in City C. This is a barrier for the operation of institutions and the families of the elderly. Therefore, I hope that the next step of LTCI can achieve cross-regional enjoyment.*
(P-1)

Secondly, the “Evaluation Criteria (Trial)” does not include evaluation indicators for dementia, such as emotional and mental state or psychological state, and the conditions for treatment are more severe. This has caused this group to be excluded from the system—a group that includes people with moderate or mild care needs who are semi-disabled and chronically ill, those with dementia, and the non-elderly disabled. In addition, the current disability assessment standards in different regions are significantly different, and this fragmented pattern will also hinder the implementation of a unified national assessment system.


*At present, there are many institutions like us in City C that takes care of all the elderly from self-care to semi-disabled and disabled. The ratio of disability to self-care among the elderly is about 1:1, which means that there are many healthy elderly or semi-disabled. Evaluating them with the same criteria is not very accurate to capture their needs sometimes, but if we use different criteria for different people, our cost is too high.*
(P-1)


*We usually use the ADL scale to assess the level of disability before the elderly are admitted to the institution. The assessment content includes the performance of various aspects such as thinking awareness, social communication, and activity ability. But there are some elderly who are demented or a little confused, and their consciousness goes up and down. When they are awake, they can dress and eat by themselves like normal people, so there is impossible to evaluate them as incapacitated at this time.*
(P-5)

#### 3.2.2. Type of Provision: The Substance of Service Supply Is Single, and the Supply Lacks Precision and Dynamic Adjustment Mechanisms

On the one hand, the service supply substance of each pilot city mainly focuses on two aspects: medical care and basic life care. Care services, such as preventive health care, rehabilitation training, psychological comfort, and end-of-life care, have been lacking for a long time. In terms of service form, except for Qingdao, where services are relatively comprehensive, most cities focus on specialized medical nursing and institutional nursing. The fundamental role of home-based and community-based care has not been adequately emphasized.


*At present, there is no problem in the centralized management of the institution for one caregiver to manage one district at night. But if you let the caregiver come to the door. How many caregivers can there be? How much can you provide them? How do the caregivers get from one house to another? What happens if a caregiver is involved in a traffic accident while on the road? How to provide in-home service when the elderly need to go to the toilet or to turn over in the middle of the night? This working time and service standards require reasonable arrangements. It is difficult for caregivers to provide in-home services now.*
(P-10)


*In the process of providing home-based care services, the biggest difficulty we encounter is the lack of cognition of the elderly. Some elderly will refuse to open the door if they disapprove of you. They think they can take good care of themselves, so they don’t need people from the institution to disturb them. Another is that it is very difficult to recruit such caregivers who provide in-home services.*
(P-8)


*I think LTC services are mainly to provide the elderly with daily care and basic medical services, while at the same time giving them psychological comfort. Medical care services, on the other hand, focus more on post-acute care and rehabilitative care to help the elderly recover their physical functions.*
(P-3)


*Some elderly who are a bit confused are not able to eat or go to the bathroom by themselves when they are not conscious. So they need caregivers to take care of them. Especially since they often engage in dangerous behaviors such as running around or hitting people, caregivers are also required to soothe them.*
(P-2)

On the other hand, a nursing-level evaluation standard and dynamic management mechanisms corresponding to disability levels have not been established. The standards set for nursing grades across institutions vary greatly and they are mainly designed to cope with inspections, resulting in the mere formality of nursing grades and mismatches between service supply substance and actual needs. At the same time, when disability levels change, nursing levels and treatments cannot be dynamically adjusted, which leads to inefficient use of nursing resources.


*We build up our own level of care based on national standards. According to the different degrees of disability of the elderly, it is divided into three levels: self-care, assistance, and nursing, and formulate corresponding nursing service plans for the elderly.*
(P-6)


*Although we are divided into nursing grades and pay different nursing fee, actually we all get the same service. In addition, they did not adjust my nursing level after I got better, and I paid a lot of nursing fee every month.*
(P-7)

#### 3.2.3. Delivery Strategy: Socialized Care Synergy Cannot Be Formed and Informal Care Lacks Policy Support

On the one hand, the main service providers in most pilot cities are designated medical and nursing institutions, while the participation of social forces, such as social organizations and home and community-based care institutions, is insufficient. Access to private for-profit service providers is difficult, and the implementation of various subsidies and policies is vague, resulting in a serious shortage of care service providers. A service delivery platform between institutions, homes, and communities has not yet been established. It is difficult to integrate care resources and achieve synergy in social care.


*We are located in a very awkward position. On the one hand, we belong to the elderly service programme added to the Chinese hospital, but for some reasons we are not recognized as a medical addition, so we did not receive operating subsidies. On the other hand, we get a business license, which means that the nature of the institution is not public-private, and thus we did not receive subsidies in the civil affairs department. I hope that the government give support to the emerging medical and nursing care projects.*
(P-8)


*In 2017, we launched a community project—signing a contract with the government to deliver meals to the home-based disabled elderly in three communities. The site provided to us by the government was a semi-basement that was very difficult to renovate. So we spent CNY 400,000 renting and renovating a house by ourselves. The government only gave a subsidy of CNY 140,000. After one year of operation, we calculated that we were losing money. I think home-based service is a very good thing and there will be great demand in the future. But not now, not for the current generation over 70. This generation is accustomed to suffer and is unwilling to bother children or spend money on services.*
(P-10)

On the other hand, current caregiving teams are far from meeting the demands of LTC services for the disabled, both in terms of quantity and quality. The existing caregiving teams are older and less well educated. After the state abolished the qualification certificate for elderly caregivers, new problems have arisen with respect to access, training, and certification. The shortage of professional caregivers has led to the emergence of an alternative role for informal caregivers [33]. However, the mode of care for the disabled elderly in China is undergoing a transition from family care to social care now [34]. Informal care is still a private sector matter that has not fully attracted the attention of policymakers. There is a gap between social and family policies to support the development of informal care.


*We have over 20 caregivers, each working 12 hours per shift. There are really too few caregivers and too much turnover. The work is tiring and the money is low, so no one wants to be a caregiver anymore. We have been having various training, but after the state abolished the nursing caregiver qualification last year, it is not clear how to go next.*
(P-5)


*I can’t move my leg after a fall. After my partner left, my daughter was worried that I would be at risk at home alone. She was too busy to take care of me, so she sent me to this nursing home. To be honest, the conditions here are average. If I have the conditions, I want to be at home absolutely. However, I don’t want to burden my children.*
(P-9)


*I have two sons who usually visit me at night after work. They don’t know how to do services like turning or bathing, and they are not as clean and comfortable as the caregivers. That’s why it’s good for me to live here.*
(P-4)

#### 3.2.4. Finance Mode: Over-Reliance on Medical Insurance Funds and Unfair Financing Standards

On the one hand, in practice, the pilot cities still use the transfer of the medical insurance pooling fund and the adjustment of the medical insurance unified account structure as the main source of funds. In some pilot cities that have explored individual contributions or organizational contributions, the responsibility for payment by individuals or organizations has not really been implemented. It has become an indisputable fact that LTCI is subordinate to basic medical insurance in actual operations [35], which violates the separate establishment principle of “sixth insurance”. Payment level and coverage are restricted by basic medical insurance. When the balance of medical insurance funds is insufficient, LTCI will not be able to continue to operate.


*The central government’s pilot guidance in 2016 proposed that in the pilot stage, funds can be raised by optimizing the structure of the medical insurance unified account and transferring the balance of the medical insurance overall fund. We implemented it with reference to the documents, and guaranteed the payment at the beginning of the establishment of the LTCI system. However, affected by the ageing of the population, the ageing degree of the population in J province is higher than that of the whole country, if funds continue to be derived from the medical insurance fund in the future, it will bring great pressure to the operation of the medical insurance fund and even the entire medical insurance system.*
(P-11)

On the other hand, at this stage, the funding criteria in the LTCI pilot vary with respect to many factors, such as geography, occupation, and age. The huge differences in funding will definitely lead to differences in levels of protection and the quality of services. This will lead to new system fragmentation and social injustice, hindering the full establishment of an adequate system.


*Our northeastern provinces have a relatively low level of economic development, so the funding level and treatment level is certainly not as good as the developed eastern provinces. Therefore, the pilot is run according to the actual situation, but at the same time, we should also work towards the national unification. For example, the current funding level of C city can basically maintain the operation of the system of LTCI. However, if the coverage is expanded in the future, it will cause dissatisfaction among different groups internally and be too far away from other provinces and cities externally. The system will not be sustained.*
(P-12)

#### 3.2.5. External Environment: Economic and Social Development Makes It Difficult to Cope with Ageing Needs and Uncertain Risks

China’s ageing population has entered a stage of rapid development, in which the elderly and disabled co-exist. The LTC guarantee mechanism and service supply system have not yet been established, leading to a lack of recourse for those in need of care. Economic growth has entered a new normal. Affected by the downward pressure on the economy and the COVID-19 epidemic, on the one hand, enterprises are calling for tax cuts and fee reductions, which have weakened public fiscal revenue and livelihood expenditure. On the other hand, the increase in public health expenses has brought greater pressure on the supply of medical insurance funds, affecting the fundraising and sustainable development of the system.


*The COVID-19 epidemic has severely impacted economic development. Business shutdowns and increased unemployment have affected the ability and stability of payment. Associated with the increase in expenses for public health prevention, the burden on the medical insurance fund has increased. It has become very difficult to raise money for LTCI.*
(P-12)


*Because there have been cases of group infection of elderly in nursing homes in other countries, we pay more attention to the closed management of senior care institutions. As a result, senior care institutions cannot provide home-based care services, and the elderly at home cannot enjoy the protection of LTCI. The expansion of LTCI is not ideal.*
(P-11)

## 4. Discussion

Through a survey of four LTCI pilot cities in three northeastern provinces, we found that pilot policies and practices are beset by problems of insufficiency, unfairness, unsustainability, and fragmentation The cases of these pilot cities are not exceptional. The degree of population ageing in Northeast China is higher than elsewhere in the nation. Economic development is falling behind, and the supply capacity of care services is seriously insufficient. The problems existing in the northeastern LTCI pilot programmes, which are universal and representative, reflect the dilemma that confronts system exploration in China. Therefore, in the context of the advent of an ageing society and an active response to population ageing becoming a matter of national strategy, we have taken a comprehensive approach to assess the overall situation. On the basis of summarizing the pilot experience of the three northeastern provinces and fully absorbing the lessons from the reforms of typical developed countries [13,36,37], we have formulated a unified top-level plan for the nation that is conducive to promoting the rational establishment and steady progress of the LTCI system.

### 4.1. Further Expand Pilot Coverage and Optimize Disability Assessment Criteria in Due Course

On the one hand, limiting the coverage of the LTCI system would violate the principle of equity and affect the scale of funding (c.f., Japan, South Korea). Therefore, China’s LTCI should clarify the concept of fairness for universal coverage. In view of the current pilot status, we should further expand policy coverage by requiring the equalization of basic public services. The pilot phase takes the key target group as the insured. With the expansion of the pilot and the finalization of the system, all urban and rural residents should be included. At the same time, there should be a focus on the rights and interests of new types of workers, flexible employees, migrant workers, and other special groups to ensure their participation in the insurance scheme. Ultimately, a universal care insurance system covering all people, with equal participation and moderate protection, will be formed. On the other hand, payment conditions should be designed objectively and meticulously to avoid the problem of welfare-dependence as a result of the generosity of benefits. The conditions for payment should be gradually relaxed and the scope of reimbursement should be extended. Each pilot city should dynamically adjust its assessment standards. A departmental collaboration mechanism and am information linkage mechanism should be established as soon as possible to strengthen the coordination and precise articulation of current assessment standards in accordance with national standards. In addition, a dynamic adjustment mechanism should be established to ensure the scientific and timely nature of disability assessment. The top-level collaborative design of the dementia and disablement assessment system should also be strengthened, so that more groups with dementia can enjoy care protection.

### 4.2. Enrich the Substance of Service Supply according to Needs and Develop Scientific Nursing Grade Evaluation Standards

Countries such as the Netherlands have been forced to cut back on formal care under financial pressure, resulting in a severe shortage of care supply [10]. In order to fully meet the LTC needs of the disabled and maintain their quality of life and dignity as much as possible, China should include a series of LTC services, such as rehabilitation exercise, psychological comfort, and end-of-life care in the scope of protection, and enhance the level of policy protection, with the needs of the disabled elderly at the core. Secondly, scientific and reasonable care rating standards and assessment systems should be developed and specific service payment catalogs designed. Thirdly, the promotion of a nationwide unified and interconnected information platform and a basic database for care services ought to be speeded up. We should carry out dynamic health monitoring to explore the LTC needs of the disabled and dementia groups over time so as to match supply and demand effectively.

### 4.3. Accelerate the Cultivation of the Socialized Service Supply Market and Build a Multi-Service Supply System

It has gradually become an international consensus that home-based care plays an important role in LTC services. However, due to unclear policy orientations and imperfect informal care support policies in typical countries, the fundamental role of family and community care cannot be fully realized. At present, the pilot cities in China are exploring the establishment of a home bed care model. To establish a “15 min elderly care service circle”, we should make full use of the specialized advantages of medical and nursing institutions to enrich the modes of service supply according to need. The goal is to eventually form a community and home care service system that is based on home care beds and centred on home-based care. We should increase government policy guidance and financial support, guiding social forces to take part in the delivery of care services by purchasing services and issuing service vouchers and family hospital beds. At the same time, informal caregiver support policies should be established and informal caregivers, such as family members or community volunteers, should be encouraged to provide care for the disabled and demented through cash subsidies or welfare payments (e.g., flexible work, paid care leave for one child, time banks). Regular professional training and counseling to improve the professionalism of informal care are also essential.

### 4.4. Build a Multi-Channel Independent Financing Mechanism with Mutual Help and Shared Responsibility

In the context of serious ageing, relying on the medical insurance fund to run the LTCI system would cause a funding deficit, which would affect the sustainability of the system. The sustainability of funding sources depends on diversified funding channels, a scientific and reasonable funding structure, and the implementation of funding responsibilities. In the future, with the aim of building a multi-channel independent financing mechanism with mutual assistance and shared responsibility, we should clarify the institutional responsibilities of basic medical insurance and LTCI and integrate LTCI with basic elderly services. On the one hand, the share of medical insurance funds in financing should be gradually reduced. The funds should be maintained in separate accounts and accounted for separately. On the other hand, the principle of linking rights and obligations should be consolidated, and the awareness of individuals and organizations when it comes to paying contributions and fulfilling responsibilities should be heightened. At the same time, taking into account the degree of ageing in each region, the level of disposable income of residents, and other factors, the government should increase financial subsidies for less developed regions and economically disadvantaged groups.

### 4.5. Establish a Systemic View to Integrate LTCI Policy into National Overall Planning and Consider It in an Integrated Manner with Other Economic and Social Policies

General Secretary Xi Jinping emphasized during the 28th collective study of the Political Bureau of the CPC Central Committee that “It is necessary to accurately grasp the connection between various aspects of social security, the social security field and other related areas of reform. Improve overall planning and coordination and promotion capabilities.” The operation and development of the welfare system cannot be separated from the economic and social environment. The design of the welfare system should be based on full consideration of external environmental factors. During the operation of the system, we should always pay attention to changes in the external environment and strive to create a suitable operating environment. Therefore, the LTCI system should be considered together with other economic and social systems. The whole process of system design, pilot programmes, expansion, and establishment should highlight the systemic integration of economic, social, technological and cultural developments.

## 5. Conclusions

This study has taken the pilot practices of LTCI in three northeastern provinces as an entry point, developed an analytical framework for social welfare policy, evaluated the effects and problems of the pilot implementations of LTCI policy in China along five dimensions, and proposed optimization options. Based on in-depth interview analysis, our study shows that there is still a large discrepancy between pilot practice and goals. Specifically, the allocation base is based on the principle of “selective allocation + diagnostic differentiation”, which imposes various restrictions on participants in terms of geography, occupation, and disability status. The coverage of the system is narrow; in terms of the type of provision, limited care resources and supporting policies have resulted in a single supply of policy pilot services. It is difficult to achieve a precise connection between supply and demand; in terms of a delivery strategy, various subsidies and policy support are vague, resulting in a serious shortage of formal and informal care providers. It is difficult to achieve synergy in social care; in terms of financing modes, LTCI currently relies heavily on medical insurance funds and has different financing standards. There is a lack of fairness and integrity; in terms of the external environment, economic and social development makes it difficult to cope with the needs of ageing and uncertain risks.

Therefore, on the basis of fully absorbing the experience and lessons of typical developed countries and local pilot practices, this study puts forward the following suggestions: China’s LTCI system should be based on the premise of the independent establishment of the sixth insurance. The distribution foundation should be consolidated by expanding the coverage of the pilot programmes and optimizing disability assessment criteria; supply and demand channels should be linked by providing services on demand and formulating scientific nursing-grade evaluation standards; a multi-supply system ought to be built by accelerating the cultivation of the socialized service market; sustainable financing mechanisms should be improved based on the principles of mutual assistance and shared responsibility; a systematic view should be established to incorporate LTCI policies into a national top-level design plan, considering them in an overall manner along with other economic and social policies. While dynamically adjusting the fragmented policy differences across the region, we are promoting the full establishment of LTCI with the core aim of meeting the care needs of the disabled population.

This study is limited by two main factors. First, this study has used a qualitative research method on a small scale, exploring the common problems of LTCI policy pilots in China. Thus, the credibility of the current analysis can be bolstered by both quantitative and qualitative studies in the future. Although the sample size is small and the findings retrieved from a small number of experiences might not be generalized to every region, the study results could provide a more authentic and objective data support for future experience sharing and pilot promotion. Second, considering cross-cultural challenges, the options for optimizing LTCI policies proposed in this study may only be useful for middle-income countries in East Asia, where similar challenges are faced by families and societies to meet the health care and long-term care needs of a rapidly ageing population.

## Figures and Tables

**Table 1 ijerph-19-04298-t001:** Description of research participants.

No.	Gender	Age	City	Status	Notes
P-1	Female	33	City C, Province J	Manager	Private for-profit senior care institutions; designated institutions for LTCI in City C; mainly for the disabled, senior, and dynamic elderly
P-2	Female	46	City C, Province J	Caregiver	
P-3	Female	62	City C, Province J	Medical staff	
P-4	Male	84	City C, Province J	Elderly	
P-5	Female	42	City T, Province J	Manager	Private for-profit senior care institutions; designated institutions for LTCI in City T; mainly for the disabled and semi-disabled elderly
P-6	Female	53	City T, Province J	Caregiver	
P-7	Male	78	City T, Province J	Elderly	
P-8	Female	51	City Q, Province H	Manager	Publicly run non-profit combined medical and nursing institutions; designated institutions for LTCI in City Q; mainly for the dynamic, disabled, and semi-disabled elderly
P-9	Male	86	City Q, Province H	Elderly	
P-10	Female	54	City P, Province L	Manager	Private non-profit senior care institutions; designated institutions for LTCI in City P; mainly for dynamic, disabled, and semi-disabled elderly people
P-11	Male	28	City C, Province J	Medical Insurance Bureau staff	2 years in the industry, familiar with the implementation of LTCI policy
P-12	Male	34	City C, Province J	Medical Insurance Bureau staff	2 years in the industry, familiar with the implementation of LTCI policy

**Table 2 ijerph-19-04298-t002:** Semi-structured interview schedule.

System Elements	Respondents	Questions
Allocation base	Manager	1. Who is covered by LTCI in your institution? Do these people belong to the severely disabled elderly population? What other groups do you think could be covered by LTCI?
	Manager	2. Did the elderly receive a disability-level assessment when they checked in? What are your evaluation criteria?
Type of provision	Manager	3. What difficulties do you see in the delivery of institutional services in community and home-based settings? What are the reasons for these difficulties?
	Medical staff	4. What do you think are the main elements of LTC services? How is LTC different from other health care services?
	Caregiver	5. In addition to the existing care services they receive, what do you think are the LTC services that the elderly really need?
	Caregiver	6. Does your institution classify the level of care? What is the basis for the classification?
	Elderly individual	
Delivery strategy	Manager	7. What do you think are the problems or difficulties with respect to government purchases?
	Manager	8. How is the allocation of caregivers in your institution decided?
	Elderly individual	9. What difficulties did your family, relatives, and friends have in taking care of you before you entered the nursing home? Would you like to go home for services if possible?
finance mode	Medical Insurance Bureau staff	10. If LTCI continues to be transferred to the health insurance fund, what impact do you think this will have on the operation of the health insurance fund?
	Medical Insurance Bureau staff	11. Do you think the current system pilot scheme should aim at national standardization or should it seek to highlight geographical differences?
	Medical Insurance Bureau staff	12. How do you think the current economic and social environment has influenced the development of the LTCI system?

## Data Availability

The data presented in this study are available on request from the corresponding author. The data are not publicly available due to restrictions of privacy.

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
