# Peer review of "Long-Term Care Insurance Pilot Programme in China: Policy Evaluation and Optimization Options—Taking the Pilot Programme in the Northeast of China as an Example"

_ijerph, 2022, doi:10.3390/ijerph19074298_

Round 1

Reviewer 1 Report

  1. What is the main question addressed by the research?

The paper focuses on a study of a long-term care insurance (LTCI) pilot programme in three north-eastern Chinese provinces. The authors analyse the general state and common features of LTCI pilot programmes in five dimensions: allocation base, type of provision, delivery strategy, finance mode, and external environment.

  1. Do you consider the topic original or relevant in the field? Does it
    address a specific gap in the field?

The process of population ageing has social and biological forms and causes that have not been sufficiently studied by contemporary socio-economic science. In this context, the paper dealing with institutional long-term care insurance is relevant.

  1. What does it add to the subject area compared with other published material?

An attempt has been made to optimize the system of long-term care, which is the main focus of countries with a high proportion of aging populations. The authors argue that China has always had a tradition of testing the rationality, feasibility, and scientificity of policy plans through small-scale pilot programs before introducing a formal system.

  1. What specific improvements should the authors consider regarding the methodology?  What further controls should be considered?

This investigation uses in-depth interviews to gather the real feelings and behavioral choices of LTCI policy participants. One nursing home in each of the four LTCI pilot cities in three northeastern Chinese provinces was selected.  Eleven participants, including 4 managers, 2 caregivers, 1 medical staff, and 3 seniors with disabilities in total participated in the study.

In our opinion, the above-mentioned sample is questionable for such an extensive research assignment.

The paper is a promising study, but it lacks more sophisticated research methods. Basing all observations only on in-depth interviews raises doubts about the quality of the scientific analysis.

  1. Are the conclusions consistent with the evidence and arguments
    presented and do they address the main question posed?

The goal of the study is to build a framework of LTCI system that adapts to the level of China's economic development and to the trend of ageing in the new development pattern, and eventually to establish a sound multi-level LTC protection system that meets the diversified needs of the public.

The analysis of the interviews only made it possible to outline the existing problems in the five dimensions stated by the authors and to state the existence of "a large deviation between the pilot practice and the goals," as well as to put forward a few suggestions to improve the functioning of the Chinese LTCI system.

  1. Are the references appropriate?

The literature cited in the paper corresponds to the current state of research on the issues discussed, however, it is worth broadening the review of publications by analyzing the views of European economists in the field under study.

  1. Please include any assitional comments on the tables and figures.

No critical remarks regarding tables and figures.

  1. Additional comments or continuations of above sections.

The research methods (in the text and in the abstract) were not clear enough.

There were generalized conclusions that were drawn from isolated judgments from the interviews.

The study would have benefited if it had demonstrated how the problems mentioned by the authors about the sustainability of the LTCI system, which are still occurring in the implementation of the system in Germany, Japan, the Netherlands and South Korea, were taken into account in the present study (line 40-89).

Reviewer 2 Report

  1. It would be helpful for readers to summarize in a chart the relations among the variables you consider. Which are the dependent and independent variables?
  2. The index appears too much detailed to me and can confuse the reader.
  3. My background in healthcare management suggests that you can dedicate some lines to the managerial aspect of the LTCI.
  4. Anyway, I really appreciate your paper.
